# Heat Stroke Prevention in Hot Specific Occupational Environment Enhanced by Supervised Machine Learning with Personalized Vital Signs

**DOI:** 10.3390/s22010395

**Published:** 2022-01-05

**Authors:** Takunori Shimazaki, Daisuke Anzai, Kenta Watanabe, Atsushi Nakajima, Mitsuhiro Fukuda, Shingo Ata

**Affiliations:** 1Department of Clinical Engineering, Faculty of Health Care Sciences, Jikei University of Health Care Sciences, Osaka 532-0003, Japan; t-shimazaki@juhs.ac.jp; 2Graduate School of Engineering, Osaka City University, Osaka 558-8585, Japan; ata@osaka-cu.ac.jp; 3Graduate School of Engineering, Nagoya Institute of Technology, Naogya 466-8555, Japan; 4West Japan Railway Company, Osaka 564-0021, Japan; kenta-watanabe@westjr.co.jp (K.W.); atsushi-nakajima@westjr.co.jp (A.N.); 5Mets Inc., Tokyo 120-0036, Japan; hiro.f@mets-tokyo.jp

**Keywords:** vital sensing, WBGT, heat stroke prevention, supervised machine learning

## Abstract

Recently, wet-bulb globe temperature (WBGT) has attracted a lot of attention as a useful index for measuring heat strokes even when core body temperature cannot be available for the prevention. However, because the WBGT is only valid in the vicinity of the WBGT meter, the actual ambient heat could be different even in the same room owing to ventilation, clothes, and body size, especially in hot specific occupational environments. To realize reliable heat stroke prevention in hot working places, we proposed a new personalized vital sign index, which is combined with several types of vital data, including the personalized heat strain temperature (pHST) index based on the temperature/humidity measurement to adjust the WBGT at the individual level. In this study, a wearable device was equipped with the proposed pHST meter, a heart rate monitor, and an accelerometer. Additionally, supervised machine learning based on the proposed personalized vital index was introduced to improve the prevention accuracy. Our developed system with the proposed vital sign index achieved a prevention accuracy of 85.2% in a hot occupational experiment in the summer season, where the true positive rate and true negative rate were 96.3% and 83.7%, respectively.

## 1. Introduction

Homeostasis of body temperature, which consists of heat production and heat dissipation, is maintained by thermoregulation. Heat stroke occurs when this balance is disrupted, starting with initial symptoms such as weakness, dizziness, and headache, and ending with death due to central nervous system disorders such as convulsions and coma. It is an important concern, especially for workers in hot specific environments such as warehouses and boiler rooms. For example, the United States Department of Labor has recommended that workers in such environments measure their wet-bulb globe temperature (WBGT) [1,2], in addition to hydrating well, and resting regularly [3]. However, no fundamental solution to avoid heat stroke in hot specific working environments has yet been addressed. For example, various methods have been proposed in existing studies, such as applications based on the WBGT [4], universal thermal climate index [5], heat flow compensation method [6], and dual heat flow method [7]. However, such methods based on deep body temperature should be invalid for workers in hot specific places. Therefore, this study focuses on the challenges in developing a heat stroke prevention system to protect hot-weather workers.

Particularly, the WBGT index is an important index for heat stroke prevention where core body temperature cannot be used for accurate prevention. It has been internationally standardized as ISO7243 by the International Organization for Standardization [8] and JIS Z 8504 in Japan [9]. The Ministry of the Environment in Japan operates a web-based system [10] for viewing WBGT data for each region. In another example, Kaiho et al. [11] attempted to calculate the WBGT using only the black-bulb temperature measured with a portable sensor, as opposed to the WBGT calculation that requires black-bulb temperature, dry-bulb temperature, and wet-bulb temperature. In addition, McCann et al. [12] reported that WBGT can improve the performance of long-distance runners based on the effectiveness of the heat stroke guidelines of the National Collegiate Athletic Association. In European countries, WBGT has been widely used in the development and operation of occupational heat health warning systems [2,13].

Since temperature, humidity, and heat radiation have a great influence on the perceived heat actually felt by a person (referred to as *perceived temperature* in this paper), the perceived temperature should differ greatly between working near and away from a heat source such as a boiler, even in the same indoor working environment. This means that the WBGT is valid only in the area close to the WBGT meter. Hence, although only a single WBGT meter is currently employed in indoor rooms in many cases, it cannot fundamentally solve the problem of different perceived temperatures. To solve this problem, it is necessary to measure the body temperature data for each individual worker using a wearable device.

In addition to the heterogeneous temperature and humidity distribution in a hot working place, it is important to consider the effects of clothing adjustment values, work intensity, and heat acclimatization [14], which are included in the latest version of the WBGT index [8]. However, in a real situation, the WBGT index may not possibly reflect the perceived temperature. For example, the temperatures experienced by skilled and non-skilled workers should differ because skilled workers may be accustomed to working in hot environments [15]. Therefore, to achieve accurate heat stroke prevention, it is necessary to investigate a personalized vital sign that can reflect the effects of these individual differences. In this paper, we propose a personalized heat strain temperature (pHST) index to adjust for individual differences in WBGT measurements, and then construct a new personalized vital sign index with pHST, heart rate, and activity indexes.

Furthermore, it is important to determine a threshold value for the proposed pHST to estimate heat strain for supporting proper heat stroke prevention. However, in hot specific environments, it is quite difficult to determine the threshold value even with personalized vital signs because experimental validation must be conducted in a dangerous situation with high temperature and high humidity for a long time, which is impossible for ethical reasons. Instead of threshold-based heat stroke prevention, this paper aims to introduce supervised machine learning for heat stroke prevention, in which a pHST meter, heart rate monitor, and an activity meter are all mounted on a wearable device. To apply supervised machine learning to heat stroke prevention, it is necessary to annotate the relevance of vital data to the degree of heat stroke. To realize automatic annotation of the supervised data, we constructed a web survey system to avoid undesirable premeasurement for machine learning. Then, we investigated several algorithms of supervised machine learning for heat stroke prevention, and evaluated the prevention accuracy based on the personalized vital sign index measured in a hot working environment.

## 2. Heat Stroke Prevention Method with Personalized Vital Signs

### 2.1. Proposed pHST Index

Recently, the WBGT index was adopted by the International Organization for Standardization (ISO) [8], American Conference of Governmental Industrial Hygienists (ACGIH) [16], and other countries because it is a quantification of the temperature experienced by a person, incorporating factors such as air temperature, humidity, and radiant heat, which have a significant effect on the heat balance of the human body. Workers in hot specific working environments are required to drink water or rest when the WBGT index approaches a dangerous temperature (for example, 31 °C of the WBGT index [17]). In reality, the WBGT index does not work well in many cases because it is limited to the location where the WBGT meter is installed, and hence, the WBGT meter location should be correctly selected. However, there is a concern requiring many WBGT meters for appropriate measurement location selection, which may be difficult in a real scenario.

For this reason, we developed a pHST meter that can measure the perceived temperature while reflecting the heat environment, clothing, and body size, as shown in Figure 1. The principle of the pHST index is illustrated in Figure 2. The pHST index consists of a thermopile sensor on the circuit board and another humidity sensor on the opposite side. The thermopile measures the radiant heat emitted from the body surface in a non-contact manner, and the humidity sensor measures the humidity inside the clothes. As shown in Equation (Equation 1), the WBGT is calculated by the wet-bulb temperature $T_{w}$ (degree), black-bulb temperature $T_{g}$ (degree), and dry-bulb temperature $T_{a}$ (degree) [1]:(1)$$\mathrm{WBGT}=0.7\times T_{w}+0.2\times T_{g}+0.1\times T_{a}.$$However, the wet-bulb temperature requires a wet-bulb thermometer, which is difficult to install in a wearable terminal. Hence, Ono et al. modified the WBGT index using a hygrometer, instead of the wet bulb and black thermometers, $T_{w}$ and $T_{g}$ as [18]:(2)$$\begin{array}{cc} {\mathrm{WBGT}}_{modified}= & 0.735\times T_{a}+0.0374\times RH\\ \; & +0.00292\times T_{a}\times RH\\ \; & +7.619\times SR-4.557\times SR^{2}\\ \; & -0.0572\times WS-4.064 \end{array}$$
where RH, SR, and WS denote relative humidity (%), total solar radiation ($\mathrm{kW}/m^{2}$), and average wind speed (m/s), respectively. Assuming a case of heat stroke prevention with the temperature measured under clothes shown in Figure 2, the total solar radiation and average wind speed do not significantly affect the proposed pHST index. Therefore, by excluding both SR and WS, we can represent the pHST index by approximating the modified WBGT index as:(3)$$\begin{array}{cc} \mathrm{pHST}= & 0.735\times T_{m}+0.0374\times RH\\ \; & +0.00292\times T_{m}\times RH. \end{array}$$

Here, $T_{m}$ is a dry-bulb temperature measured under the clothes by the developed wearable sensor. It should be noted that the proposed pHST index can efficiently reflect *perceived temperature*, namely the temperature perceived by the individual, with the combination of only the dry bulb temperature $T_{m}$ and the relative humidity RH measured by the hygrometer.

### 2.2. Heart Rate Detection with Canceling Motion Artifact

It is well known that vital data can be collected through wireless communication technologies [19]. In particular, real-time heart rate monitoring is important because heart rate increases sharply to exercise and an increase in external temperature. Additionally, tachycardia accompanies a moderate heat stroke [20] therefore, heart rate is useful for accurately preventing heat stroke. However, due to motion artifacts (MA) caused by body movement, it is difficult to precisely measure the heart rate when a person is working. To accurately measure the heart rate of workers, we previously developed a heart rate detection system with effective body motion artifact cancellation [21,22]. Figure 3 shows the principle of the MA cancellation; two pulse wave meters are placed in close proximity to each other, with one pulse wave meter directly touching the skin, and used as a normal heart rate monitor based on photoplethysmography (PPG) (here, this pulse wave meter is called the *PPG sensor*). The other is placed floating from the skin to detect MA (herein called the *MA sensor*). In the absence of body movement (i.e., during rest), we can measure the pulse wave waveform without MA noise at the PPG sensor. On the other hand, in the case of body movement (i.e., during exercise and working), the pulse wave superimposed by the MA noise is observed at the PPG sensor however, the MA sensor can detect only the MA noise. A recursive adaptive filter is applied to extract only the pulse wave based on the difference between the data measured by the PPG and MA sensors, so that the heart rate can be accurately obtained even during exercise and working.

### 2.3. Activity Amount

The risk of heat stroke varies with the amount of activity even in the same heat environment. Therefore, we introduce a three-axis acceleration sensor as an activity meter to measure the activity amount. Assuming the acceleration vectors at the three-dimensional axis at time *t* and the number of samplings as $a_{t,xyz}={\left[a_{t,x},a_{t,y},a_{t,z}\right]}^{T}$ and *n*, respectively, the activity amount can be defined as the sum of the norms by:(4)$$I_{xyz}=\sum_{i=1}^{n}\sqrt{a_{t_{i},x}^{2}+a_{t_{i},y}^{2}+a_{t_{i},z}^{2}}.$$

### 2.4. Web Survey-Based Automatic Annotation for Supervised Machine Learning

To correctly assess the risk of heat stroke, it is important to develop a decision system based on the measurement of several kinds of vital data, such as the pHST, heart rate, and activity amount. In this study, we aimed to apply supervised machine learning to heat stroke prevention. In supervised machine learning, annotation (labeling) of measured vital data is necessary to obtain the relevance of the vital data to the degree of heat stroke, thus classifying the subjects into non-thermal and thermal groups. Automatic annotation to prepare supervised data can avoid undesirable premeasurement for supervised machine learning. Hence, we developed a web-based survey system to realize automatic annotation. By answering the survey on the web before and after working with a smartphone or laptop PC, the measured vital data can be automatically annotated to their daily physical condition and biometric data related to heat stroke.

Figure 4 shows the input form of the web survey for the developed automatic annotation. The web survey was conducted before and after working, and the measured vital data was automatically annotated to the degree of heat stroke based on the answers in the web survey. In the developed system, we set a border line to prevent heat stress at the 3rd answer in the after-work web survey. Selection below the 3rd answer denotes the non-heat stroke group (negative), whereas the group that chooses the 3rd answer or higher is labeled as the heat stroke group (positive).

## 3. Methods

The developed wearable device was fixed to the upper arm with a belt for 20 adult male subjects (height, weight, BMI, age, and gender were recorded) working at a train maintenance factory in Osaka, Japan. The average outdoor temperature and relative humidity of Osaka in August were around 28 °C and 72%, respectively. Biometric data was acquired from 10:00 to 17:00 (excluding lunch break and including overtime work) from Monday to Friday for the month of August. Approval for this vital data measurement experiment was obtained from the West Japan Railway Company. Note that the workers responded to the survey in the experiment, and we confirmed that they answered all survey questions properly.

Figure 5 and Figure 6 show an overview of the developed wearable device and vital data gathering system, respectively. As shown in Figure 5, the developed device consists of a PPG sensor, MA sensor (non-contact PPG sensor), three-axis accelerometer, thermopile, and humidity sensors. The heart rate and activity amount were measured at a sampling rate of 10Hz, whereas the body surface temperature, humidity, and temperature inside the clothes were acquired at a sampling frequency of 10Hz. All measurement data are stored in the random access memory (RAM) of the wearable device for 1min and then sent to a cloud server via the long-term evolution (LTE) connection illustrated in Figure 6. The supervised machine learning-based system performs heat stroke prediction in the cloud server. Table 1 summarizes the detailed specifications of the hardware configurations.

## 4. Results

As a measurement example, we demonstrate the time series of the measured biometric data, environmental temperatures, and wind speed in Figure 7. As can be seen from Figure 7a,b, the pHST was not consistent with the WBGT. This is because the WBGT reflects environmental heat stress, on the other hand, the pHST depends on microenvironment heat. Furthermore, it was confirmed that drinking water had a good effect on the heart rate and pHST. The data collecting server converts the measured vital data to into the pHST, heart rate (HR), activity amount, integrated value of each pHST (Int. pHST), integrated value of each heart rate (Int. HR), and integrated value of each activity amount (Int. activity), which are stored in the database in the cloud server and further converted to feature values for supervised machine learning.

### 4.1. Features, Classifiers, and Evaluation Indexes for Supervised Machine Learning

To select a classifier for supervised machine learning-based heat stroke prevention, we used six-dimensional features: The pHST, HR, activity amount, Int. pHST, Int. HR, and Int. activity. Candidates for the classifiers are *k*-nearest neighbor (KNN, k=5 with Euclidean distance), decision tree (maximum number of partitions =100), ensemble algorithm (Adaboost, maximum number of partitions =100), logistic regression, Bayes (Naive Bayes with Gaussian distribution), and linear support vector machine (SVM). To avoid overtraining, we employed a 10-segment cross-validation. In the experiment, we investigated an optimal classifier in terms of not only the accuracy but also the learning time. The reason for considering the learning time for classifier selection is to suppress the computational cost of the learning model, which should be updated every day to cope with sudden changes in climate such as extreme weather and early summer.

In general, accuracy, true positive rate (TPR), true negative rate (TNR), false negative rate (FNR), and false positive rate (FPR) are used as criteria for evaluating the performance of supervised machine learning, as defined by:(5)$$\begin{array}{cc} \; & \mathrm{Accuracy}=(TP+TN)/(TP+FP+TN+FN) \end{array}$$(6)$$\begin{array}{cc} \; & \mathrm{TPR}=\mathrm{Sensitivity}=TP/(TP+FN) \end{array}$$(7)$$\begin{array}{cc} \; & \mathrm{TNR}=\mathrm{Specificity}=TN/(FP+TN) \end{array}$$(8)$$\begin{array}{cc} \; & \mathrm{FNR}=FN/(TP+FN) \end{array}$$(9)$$\begin{array}{cc} \; & \mathrm{FPR}=FP/(FP+TN) \end{array}$$
where *TP*, *TN*, *FP*, and *FN* are the number of correct predictions for positive and negative subjects, and the number of incorrect predictions for positive and negative subjects, respectively. To evaluate these evaluation indexes, we divided the measurement data of each subject into training data (80%) for supervised learning and test data (20%) for the evaluation.

### 4.2. Experimental Results

First, we investigated an optimal classifier for the heat stroke prevention system. Figure 8 illustrates the learning time and accuracy for each classifier. Here, we evaluated the accuracy of the developed prevention system based on comparison with the survey results. As can be seen from Figure 8, the KNN classifier achieved the highest accuracy of 85.2%. The decision tree algorithm is a well-balanced classifier in terms of both, the accuracy and learning time. Based on the evaluation results, we selected the KNN classifier to achieve the best accuracy and acceptable learning time in terms of heat stroke prevention.

Hereafter, let us discuss the dependency of the selection of the features on prediction accuracy. Figure 9 shows the accuracy of the feature combination when the KNN classifier is adapted. It can be seen from Figure 9 that the heat stroke prediction with the combination of all features (six dimensions) achieves the highest accuracy. This means that the combination of several types of biometric data can drastically improve the accuracy. In particular, the pHST index contributes significantly to improving the prediction accuracy. Therefore, our proposed temperature index, the pHST index, can successfully represent the realistic temperature felt by each individual worker in the experiment, which should be different from the ambient environmental temperature indicated by the conventional WBGT index.

## 5. Discussion

Herein, we discuss the relationship between the measured temperature and annotation via the web survey in detail. Table 2 shows the days when WBGT and pHST reach a dangerous temperature (31 °C or higher) as announced by the Ministry of the Environment, Japan [23]. This figure also includes the results when a worker selected the 3rd answer or higher (the border line to feel heat strain) in the postwork survey during the experimental period (August, Osaka City). The number of days when the WBGT reached the dangerous temperature was 12 however, the number of days when the heat strain was indicated via the postwork survey was only 5. Additionally, no heat strain was indicated in the postwork survey despite the WBGT indicating a dangerous temperature for five consecutive days from the 27 August. This suggests that the environmental and perceived temperatures may be different from each other in the WBGT measurement under hot specific environments.

On the other hand, the number of days when pHST reached 31 °C or higher was 12. The heat strain indication results via the web survey match 10 out of 12, suggesting our proposed pHST index is a good indicator for heat stroke prevention in a heat specific working place thanks to effective measurement of the perceived temperature. The effectiveness of the pHST is also shown in Figure 9; considering only a one-dimensional feature, the pHST achieved the best accuracy of 69.5% for all features. The activity amount reached the second highest accuracy, and the accuracy with HR was the lowest. This is because the most significant factors for heat stroke are the temperature of each individual worker and the activity amount. While working in a hot occupational place, the direct causes of heat stroke are the temperature and activity but HR remains an indirect cause. As a result, when using a combination of all the features, the accuracy becomes the highest, as shown in Figure 9. Furthermore, the integrated values of the pHST, HR, and activity amount are also important information because even a small amount of heat accumulates for a long time.

Finally, Figure 10 demonstrates the confusion matrix when the KNN classifier is applied with a combination of all the features. Here, the vertical axis (true value) is the non-heat stroke group (Neg.) and heat stroke group (Pos.) based on the web survey results, and the horizontal axis shows the estimation results of our developed heat stroke prevention system. From the results, TNR = 83.7% and TPR = 96.3% are acceptable results for heat stroke prevention in hot occupational places. However, we need to further suppress the result of FNR = 3.7% because FNR is an important indicator of system reliability, which is the missing probability of workers suffering from heat stroke. One of our future subjects is to introduce another vital sign to correctly indicate subjects who are not in a good condition and to build a more suitable learning model according to their physical condition.

## 6. Conclusions

Accurate heat stroke prevention is more important in hot specific working places than in general environments. In this study, we developed a heat stroke prevention system using a wearable vital measurement device for workers in hot environments. To construct a personalized vital sign index, a combination of several types of vital signs was introduced into the prevention system. In particular, a core body temperature should be the main factor in heat stroke prevention. Hence, we proposed the idea of using the pHST based on temperature/humidity measurement to adjust the WBGT index to realistically obtain the heat perceived by each individual worker. To further improve the prevention accuracy, this study applied supervised machine learning techniques, and then investigated optimal features and classifiers to achieve high accuracy and reasonable learning time. Importantly, we also constructed an automatic annotation system with a web survey to obtain the relevance of the vital data to the degree of heat stroke, which is essential to realizing supervised machine learning-based heat stroke prevention. The evaluation results demonstrated that our developed system with the KNN classifier achieved an accuracy of 85.2%, where TPR = 96.3% and TNR = 83.7% for the preliminary heat stroke group, whose results should be applicable to heat stroke prevention in hot specific occupational environments.

Finally, the contributions of the paper are summarized as follows:This paper proposed a personalized vital sign index by combining several types of vital data, which could efficiently prevent a heat stroke for persons in a hot working place. In particular, we improved the quality of the WBGT index as a pHST to consider the perceived temperature in the heterogeneous temperature and humidity distribution.To address the difficulty of corresponding the relevance of vital data to the degree of heat stroke, an automatic annotation system was developed for realizing supervised machine learning-based heat stroke prevention.The performance of heat stroke prevention for different types of supervised machine learning algorithms was experimentally evaluated in a hot and high-humidity specific working environment: A train maintenance factory in August in Japan.

## Figures and Tables

**Figure 1 sensors-22-00395-f001:**
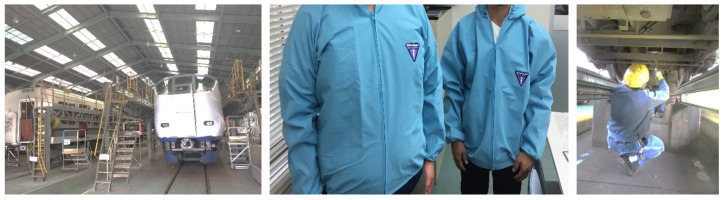
Example of hot specific working environment (train maintenance factory).

**Figure 2 sensors-22-00395-f002:**
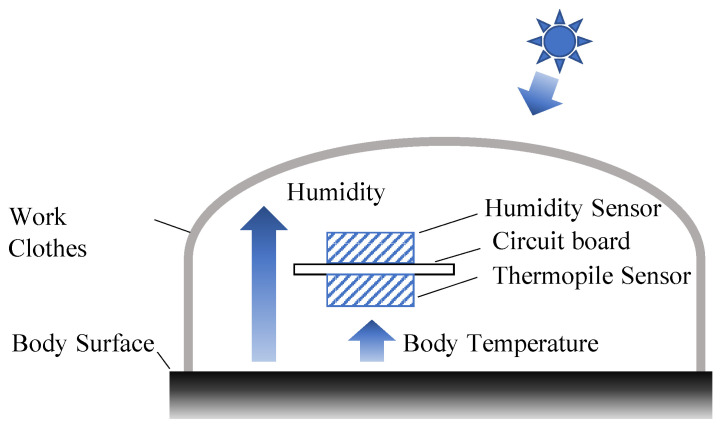
Principle of personalized heat strain temperature (pHST) measurement.

**Figure 3 sensors-22-00395-f003:**
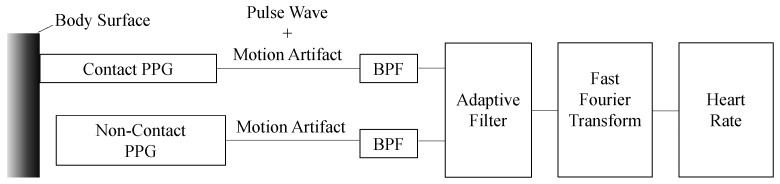
Principle of motion artifacts cancellation.

**Figure 4 sensors-22-00395-f004:**
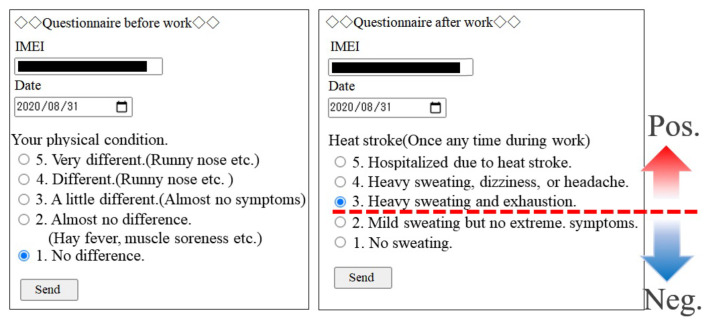
Survey form in before- and after-workings.

**Figure 5 sensors-22-00395-f005:**
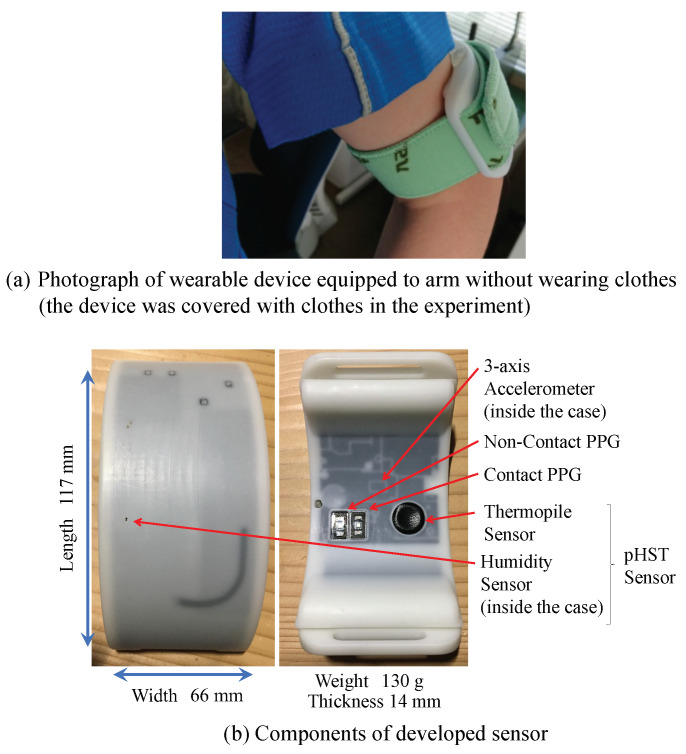
Developed wearable device for vital data sensing.

**Figure 6 sensors-22-00395-f006:**
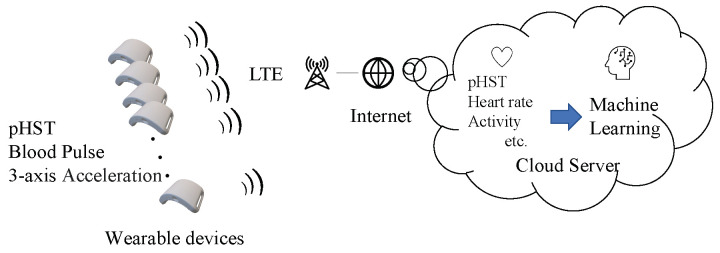
System development to collect vital sensing data.

**Figure 7 sensors-22-00395-f007:**
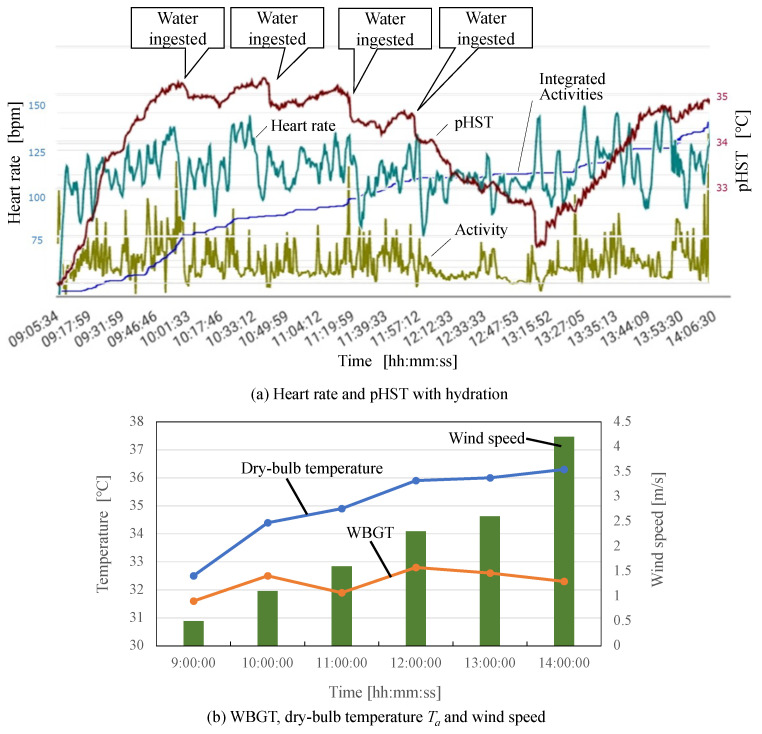
Variation of heart rate, pHST, environmental temperatures, and wind speed.

**Figure 8 sensors-22-00395-f008:**
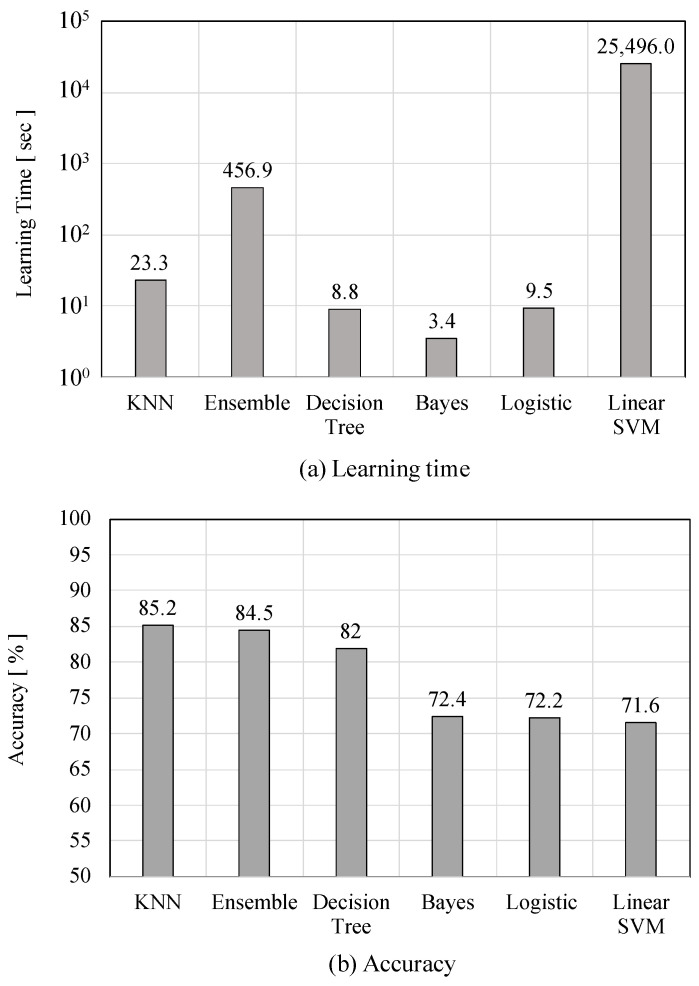
Performance evaluation for different classifiers.

**Figure 9 sensors-22-00395-f009:**
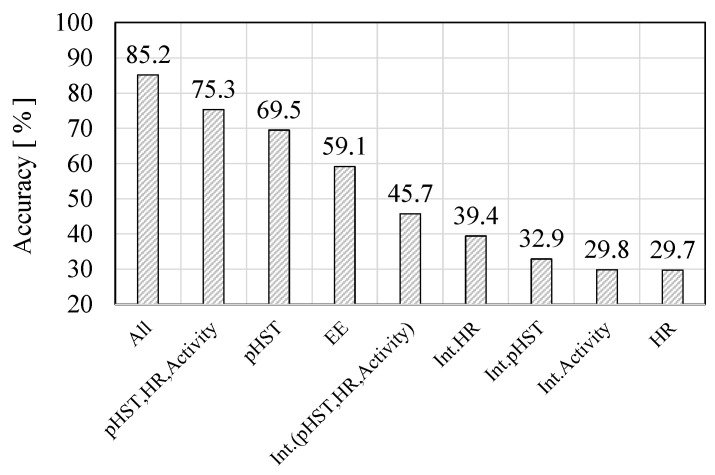
Dependency of feature selection on prevention accuracy.

**Figure 10 sensors-22-00395-f010:**
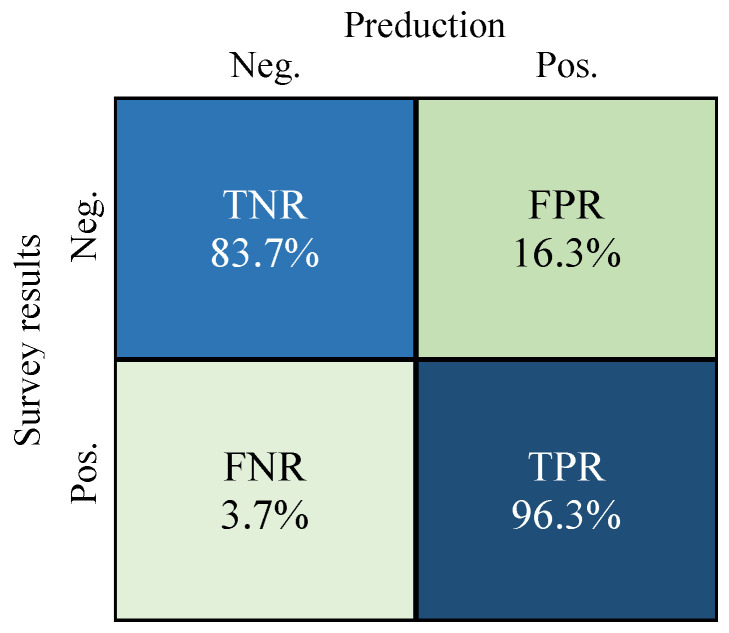
Confusion matrix for KNN classifier.

**Table 1 sensors-22-00395-t001:** Hardware configurations.

Parts	Model Number	Features
Optical sensor IC for heart rate monitor	BH1790GLC-E2	Pulse wave
Optical sensor IC for heart rate monitor	BH1790GLC-E2	Motion artifact cancellation
Thermopile	MLX90614ESF-BCC-000-TU	Body surface temperature
Humidity-temperature sensor	Si7021-A20-IM1	Humidity and temperature in clothes
Inertial measurement unit	MPU-9250	3-axis acceleration
Wireless module	4GIM V1.0	Long term evolution (LTE)
Lithium-ion battery	DTP603450	1000mAh (available for up to 10 h)

**Table 2 sensors-22-00395-t002:** Heat strain indication results based on WBGT and pHST.

	3 Aug.	4 Aug.	5 Aug.	6 Aug.	7 Aug.	10 Aug.	11 Aug.	12 Aug.
WBGT	-	-	-	-	-	indicated	indicated	indicated
pHST	indicated	indicated	indicated	indicated	indicated	-	indicated	indicated
Survey	-	indicated	indicated	indicated	-	-	-	indicated
	**14 Aug.**	**15 Aug.**	**16 Aug.**	**17 Aug.**	**18 Aug.**	**19 Aug.**	**20 Aug.**	**21 Aug.**
WBGT	indicated	indicated	indicated	indicated	-	-	indicated	indicated
pHST	-	-	-	indicated	indicated	indicated	indicated	indicated
Survey	-	-	-	indicated	indicated	indicated	indicated	indicated
	**22 Aug.**	**24 Aug.**	**25 Aug.**	**27 Aug.**	**28 Aug.**	**29 Aug.**	**30 Aug.**	**31 Aug.**
WBGT	indicated	-	indicated	indicated	indicated	indicated	indicated	indicated
pHST	-	indicated	-	-	-	-	-	-
Survey	-	indicated	-	-	-	-	-	-

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
