# Peer review of "Heat Stroke Prevention in Hot Specific Occupational Environment Enhanced by Supervised Machine Learning with Personalized Vital Signs"

_sensors, 2022, doi:10.3390/s22010395_

Round 1
Reviewer 1 Report
- The study developed a wearable device to monitor human thermal physiological responses in hot environments. Device was also used in the field by workers to compare worker’s responses with the measured pWBGT by the device. The developed device can be used to monitor human response and prevent heat related illnesses. However, the measured pWBGT and standard WBGT are two different indices and cannot be directly compared. The standard WBGT is not used for detection of heat stroke, rather it is an environmental heat stress index used to prevent heat stress. The latest version the standard has taken into account of clothing, activity intensity and heat acclimatization. The authors seem to refer to an old version of WBGT. Detail comments are provided below.
- Line 31-32, WBGT and the references values are mainly used to prevent the body core temperature rise over 38 C, therefore to prevent heat stress and heat stroke. It is not used to detect heat stroke, and it is not an indicator of heat stroke.
- Line 43-46, in the case described here, the heat stress and “perceived temperature” is not only affected by temperature and humidity, but also by heat radiation.
- Line 53-58, the latest WBGT standard includes clothing adjustment values (CAV), work intensity and heat acclimatization, which means it can take into account of different clothing, acclimatization status. Please check the latest version of WBGT (2017) and revise the manuscript accordingly. In your reference list, you used a version in 1989. This is a quite old version.
- Line 64-68, it is not clear what you meant by “optimal threshold”. Please elaborate. WBGT standard provides a table of reference values that are used as thresholds to prevent human body core temperature rise over 38 C.
- Line 78-89, consider moving this paragraph to conclusion section.
- Line 99-101, WBGT standard specifies the measurement method and calculation of mean values that should take into account changes in location, duration and activity as well as variations in time. if the measurement is not taken place in the work zone, the measurement location selection is not correct and should be changed.
- Figure 2, the measured temperature and humidity are actually the microclimate conditions between the skin and clothing. the measured value cannot be compared with WBGT. WBGT is an environmental heat stress index (not a microenvironment heat index). For example, in a thermal neutral environmental condition like an office, what you measured according to Figure 2 will be much higher than the measured value in the office environment because the microclimate temperature and humidity are much higher, air velocity is lower, etc. Therefore, what you measured is the microclimate temperature and humidity, not a standard WBGT index. Looking at figure 5, the device is worn without covering clothing. But the humidity may be somewhat influenced by sweating from the skin. The temperature and radiation sensor still measures the microclimate between the device and the skin (not environmental air temperature and air humidity). The measured temperature will be lower in very hot environments and higher in moderate and cool environments, namely the values cannot be compared with environmental WBGT index measured by the sensors that meet the standard requirements.
- Equation 2-3: I don’t think it is appropriate by directly removing radiant heat (SR) and wind speed (WS) factors from equation 2. It is mentioned in previous sections, local heat radiation may be an issue in workplaces. Even inside the workshop, air velocity is not zero.
- Line 128, section 2.4, make it clearer who responds to the survey. If workers should respond to the survey, what happens if workers have already suffered heat stroke and cannot answer the survey?
- Line 149-154, these are methods, not results.
- Figure 7, please add environmental air temperature, air humidity, air velocity, global temperature to the figure. Ideally measured WBGT using the standard method should be plotted here for comparison and validation.
- Line 198-200 and figure 8-9, it is not clear for me how the accuracy is achieved. Have you compared your measured pWBGT with recorded WBGT using standard instruments? This is one way to validate your measurements of WBGT.
- Line 203-204, the temperature felt by individual workers does not indicate heat stroke and does not have the same meaning as standard WBGT index. I agree that what you measured (pWBGT) is actually a combination of the skin temperature and microclimate humidity. Therefore it reflects the temperature felt by workers. From this point of view, I suggest you change the tile of the manuscript and focus on detection of human thermal responses and thermal strain, possibly heat stroke. Bu the device cannot accurately measure environmental WBGT as specified in the ISO standard.
- Line 212. The 3rd answer does not necessarily indicate heat stroke. Heat stroke cannot be diagnosed by the workers themselves.
- Line 210-218, in the ISO WBGT standard, if the measured WBGT is lower than the reference values, e.g. 31 C, it means that workers’ core body temperature will not rise over 38 C when working for 8 hours. This does not mean heat stroke (core body temperature > 40 C). Therefore, it is not appropriate to directly use WBGT to indicate heat stroke.
- Line 234-235, consider re-phrasing the sentence.
- Line 252-255, change “temperature on/inside a human body” to “core body temperature”. For me, pWBGT and WBGT are two different indices and cannot be compared, one is the thermal physiological responses (parameter for the skin temperature and microclimate), the other is environmental heat stress index. ISO standard WBGT is not used as an indicator of heat stroke, rather it is an environmental heat stress index used to prevent heat stress, more specifically to prevent the core body temperature from rising over 38 C (not 40 C for heat stroke). Please see my previous comments.
Reviewer 2 Report
The manuscript entitled “Heat stroke detection in hot specific occupational environment enhanced by supervised machine learning with personalized vital signs” deals with a novel measurement of heat strokes amongst workers, carried out in a specific city (Osaka, Japan) in a specific month (August). The approach consists in measurements of vital signs by a portable device, which are registered in a database, when later are used to develop a machine learning strategy to determine the possibility of a heat stroke before this happens thus helping to prevent casualties.
The topic is very interesting and the manuscript is well written. Nonetheless, my main concern is regarding the subject of humidity throughout the manuscript. This subject is not clearly mentioned, even tough has a high influence on the perceived temperature, as the authors claim in line 43.
Furthermore, authors mention that in the machine learning approach, relative humidity has low influence on the heat stroke detection: “…assuming a case inside a sauna, although both body temperature and HR should increase, the risk of heat stroke is not very high.” (line 228). This in my point of view, is very worrying, because it might give a wrong signal, believing that workers under high temperature and humidity do not have high risks of a heat stroke.
Therefore, I ask: how come the authors give a new index of thermal comfort by using the wet bulb globe temperature and at the same time claim that humidity is not that important for this new personalized index? Please state a clear and deep explanation for this.
Specific comment:
Line 89. Please give at least the average daily outdoor temperature and relative humidity of Osaka in August.
Figure 7. In the caption, there is a typo for “heart rate”.
Round 2
Reviewer 1 Report
The authors have addressed most of my commnets.
Figure 7 (b) was added to the revised manuscript. The comparison of Figure 7 (a) and (b) shows that pHST is not consistent with WBGT. One more comment: why the pHST is much lower (lower that 32 C) at about 13:15:52 when the average of the pHST is about 34.5 C and when the peak pHST is about 35 C?
Reviewer 2 Report
The comments were correctly addressed by the authors. I recommend the publication of the manuscript.
Author Response
Thank you very much for your kind review. All of authors would like to thank you for the acceptance of our manuscript.